# Clinical efficacy and safety of drug interventions for primary and secondary prevention of osteoporotic fractures in postmenopausal women: Network meta-analysis followed by factor and cluster analysis

Fei Wen[1☯], Hongheng Du[2☯], Liangliang Ding[1☯], Jinxi Hu[3], Zifeng Huang[4], Hua Huang[5], Kaikai Li[6], Yuxia Mo[7]*, Anyin Kuang[8]*

1 Department of Orthopedics, The People's Hospital of Rongchang District, Chongqing, China, 2 Department of Neurology, Yongchuan Hospital of Chongqing Medical University, Chongqing, China, 3 Department of Orthopedics, Affiliated Yueyang Hospital of Hunan Normal University, Yueyang, China, 4 Department of Orthopedics, Wu Han NO.1 Hospital, Wu Han, China, 5 Department of Neurology, Hankou Hospital of Wuhan City, Wuhan, China, 6 Department of General Medicine, The Central Hospital of Tuoshi Town, Tianmen, China, 7 Department of Gynecology, The People's Hospital of Rongchang District, Chongqing, China, 8 Department of Orthopedics, The Gaoxin District People's Hospital, Chongqing, China

☯ These authors contributed equally to this work.
* yuxia_mo@sina.com (YM); 13983131087@139.com (AK)

## Abstract

We aimed to evaluate the comparative efficacy and safety of drugs respectively for primary prevention and secondary prevention of osteoporotic fractures in postmenopausal women (PMW), and to further identify the optimal intervention(s) respectively for the two groups when efficacy and safety both considered. We searched three databases. Bayesian network meta-analyses were conducted for two efficacy outcomes (vertebral fractures and nonvertebral fractures) and two safety outcomes (tolerability and acceptability) respectively in primary prevention group and secondary prevention group. We synthesized hazard ratios (HRs) and 95% confidence intervals (CIs) for nonvertebral fractures, and risk ratios (RRs) for three others. Factor and cluster analyses on surface under the cumulative ranking curve (SUCRA) values were conducted to identify the best intervention(s) with efficacy and safety both considered. The study protocol has been registered in PROSPERO. We included 57 randomized trials involving fifteen anti-osteoporotic interventions and 106320 PMW. For primary prevention, only zoledronate (once per 18 months) reduced both vertebral (RR 0.46, 95% CI 0.28–0.74) and nonvertebral (HR 0.66, 95% CI 0.51–0.85) fractures. For secondary prevention, abaloparatide, alendronate, denosumab, lasofoxifene, risedronate, romosozumab, teriparatide, and zoledronate (once per 12 months) reduced both vertebral (RRs: from 0.17 to 0.62) and nonvertebral (HRs: from 0.54 to 0.81) fractures. PTH (1–84) and abaloparatide increased withdrawal risk. Romosozumab, teriparatide, denosumab and risedronate, with the greatest composite scores, constituted the optimal cluster having both superior efficacy and superior safety. Zoledronate used at 5 mg per 18 months, with the

**Data Availability Statement:** All relevant data are within the manuscript and its Supporting Information files.

**Funding:** No specific funding was received for this study

**Competing interests:** NO authors have competing interests.

**Abbreviations:** PMW, postmenopausal women; PMO, postmenopausal osteoporosis; PVF, prevalent vertebral fracture; BMD, bone mineral density; PTH (1–84), parathyroid hormone (1–84); RCT, randomized controlled trial; SUCRA, surface under the cumulative ranking curve; RR, risk ratio; HR, hazard ratio; CI, confidence interval; FDA, Food and Drug Administration; ACP, American College of Physicians.

similar safety as placebo, is the only drug intervention which has been shown to significantly reduce both vertebral and nonvertebral fractures for primary prevention of osteoporotic fractures in PMW; while romosozumab, teriparatide, denosumab, and risedronate are the optimal treatments for secondary prevention when efficacy and safety both considered. A limitation is that safety outcomes failed to consider the severity of adverse effects.

## Introduction

Osteoporosis, as the most common skeletal disease, leads to 1.5 million osteoporotic fractures per year in the USA, which results in a total annual cost of $20 billion [1,2]. On the other hand, osteoporotic vertebral and nonvertebral fractures can also cause kyphosis, chronic pain, a loss of self-esteem, an increased risk of death, a dependent living situation, and poor quality of life [3]. Postmenopausal osteoporosis (PMO), as the most common form of osteoporosis, accompanies a substantial proportion of postmenopausal women (PMW) and advances with aging [4]. The number of women at risk for osteoporotic fractures and the relevant medical costs in the next decades are expected to continue to rise, due to the fact that life expectancy of women in Western countries is presently over 80 years and continually grows [4]. Thus, selecting efficacious and safe anti-osteoporotic agents to prevent osteoporotic fractures in PMW is more than important.

Although osteoporosis or osteopenia (low bone density) is defined by bone mineral density (BMD) T score measured via dual energy X-ray absorptiometry (DXA) (namely, a BMD T score at or below -2.5 indicates the former, and T score between -1 and -2.5 indicates the latter) [5], BMD is just one of various important risk factors for fragility fractures, of which the vast majority occur in osteopenic individuals (i.e., those with a BMD T-score between -1 and -2.5) [6–9]. It reveals the importance of primary prevention of osteoporotic fractures in PMW. Meanwhile, women who have developed a fragility fracture are at higher risk for secondary fractures [10–12], and prevalent vertebral fracture (PVF) is a strong factor to predict future fractures [3], which reveals the importance of secondary prevention of osteoporotic fractures in PMW. Therefore, it is meaningful for clinicians and susceptible individuals to know the optimal drug intervention(s) respectively for primary prevention and for secondary prevention of osteoporotic fractures in PMW when efficacy and safety both considered. However, the 2017 American College of Physicians (ACP) guideline [13] shows that the comparative effectiveness of different osteoporosis drugs for PMW with osteopenia or osteoporosis is unknown. Meanwhile, evidence on comparative safety in the guideline [13] is also lacking.

Due to the lack of a randomized controlled trial (RCT) comparing all drug interventions for PMO, the best way of assessing their comparative effectiveness and safety is to perform network meta-analysis using the data deriving from RCTs in which an active drug intervention was compared with placebo or active interventions were compared among them. Although there are several network meta-analyses [14–21] having reported the comparative anti-fracture efficacy of some drug treatments for PMO, none of these studies provided evidence on comparative efficacy respectively for primary prevention and for secondary prevention of PMO. Meanwhile, none of them evaluated a novel drug intervention of zoledronate intravenously used at a dosage of 5 mg per 18 months [22], while none but two [14,15] evaluated two newly approved agents of romosozumab [23] and abaloparatide [24]. Furthermore, none of them performed relevant analyses to identify the optimal drug treatment(s) when both efficacy and safety considered, while none but two [16,18] performed analysis on safety outcome.

Hence, we carried out this network meta-analysis to assess the comparative efficacy and safety of fifteen anti-osteoporotic drug interventions for PMO respectively in the primary prevention group and in the secondary prevention group using vertebral fractures and nonvertebral fractures as efficacy outcomes and using tolerability and acceptability as safety outcomes, and when possible conducted factor and cluster analysis after network meta-analysis to identify the optimal treatment(s) with both efficacy and safety considered respectively for primary prevention and for secondary prevention of osteoporotic fractures in PMW.

## Methods

This network meta-analysis was carried out according to the PRISMA extension statement for network meta-analysis [25], and its protocol has been registered in Research Registry (registration number: reviewregistry749) and in PROSPERO (registration number: CRD42019139165), and is available in protocols.io (http://dx.doi.org/10.17504/protocols.io.bf9zjr76). The PRISMA checklist for this study is provided in S1 Appendix.

### Search strategy

We searched through 1 February 2019 for English articles reporting RCTs which assessed the anti-fracture efficacy and/or safety of drug interventions for primary osteoporosis in PMW via the Cochrane Library, PubMed and Embase without sample size restrictions. The second search was done on 17 April 2019. The search strategy for this study was developed using text words (e.g., "postmenopausal osteoporosis", "postmenopausal osteopenia", "postmenopausal women", "osteopenia", "osteoporosis", "romosozumab", "zoledronate", "Parathyroid Hormone (1–84)", "Denosumab", "Teriparatide", "abaloparatide", "Bone Loss*", "fracture*", "controlled clinical trial", and "randomized controlled trial"), and medical subject headings (MeSHs; e.g., "Osteoporosis, Postmenopausal", "Osteoporosis", "teriparatide", "Alendronate", "Risedronic Acid", and "Raloxifene Hydrochloride"), and medical supplementary concepts (e.g., "AMG 785", "Abaloparatide", "bazedoxifene", "Lasofoxifene", and "strontium ranelate") used as substitutes for MeSHs when appropriate. S2 Appendix provides the full search strategy. Moreover, the relevant articles referred in meta-analyses previously published were assessed for eligibility and Google Scholar was also searched for related RCTs.

### Inclusion and exclusion criteria

Four outcomes of interest for this study were efficacy endpoints (vertebral fractures, and non-vertebral fractures), and safety endpoints (tolerability, namely, withdrawals due to adverse events; and acceptability, namely, study discontinuation due to any cause). Fifteen active drug interventions of interest were abaloparatide (subcutaneous, 80 μg/day), teriparatide (subcutaneous, 20 μg/day), romosozumab (subcutaneous, 210 mg/month), denosumab (subcutaneous, 60 mg/6 months), parathyroid hormone (1–84) [subcutaneous, 100 μg/day; referred to as PTH (1–84)], zoledronate (intravenous, 5 mg, once per 24 months), zoledronate (intravenous, 5 mg, once per 18 months), zoledronate (intravenous, 5 mg, once per 12 months), strontium ranelate (oral, 2 g/day), lasofoxifene (oral, 0.5 mg/day), bazedoxifene (oral, 20 mg/day), raloxifene (oral, 60 mg/day), risedronate (oral, 35 mg/week or 5 mg/day), alendronate (oral, 35 or 70 mg/week, or 5 or 10 mg/day), and ibandronate (oral, 150 mg/month).

Eligible studies included for this study were RCTs: ① which lasted one or more than one year; ② in which participants were PMW with osteoporosis or osteopenia; ③ which compared at least one active intervention with placebo, or compared active interventions of interest among them; ④ in which women in each study arm were given an adjuvant therapy of vitamin D and/or calcium; and ⑤ which assessed one or more than one of the four outcomes

of interest. Studies, in which participants contained PMW with secondary osteoporosis, most participants were Asians, the Caucasian proportion of participants was below 50%, or the identical data analyzed in previously-published articles were reanalyzed, were excluded.

We defined those included trials in which each participant had a BMD T score of between -1 and -2.5 and didn't experience an osteoporotic fracture as primary prevention trials, and meanwhile defined those trials in which each participant had a BMD T score of below -2.5 and/or experienced at least an osteoporotic fracture as secondary prevention trials. If those trials which enrolled both women with prevalent osteoporotic fractures and women without prevalent osteoporotic fractures, we would try to search for and include the corresponding papers in which subgroup analyses were performed stratified by women with or without prevalent fractures. If those papers with corresponding subgroup analyses done were available, in them the data deriving from women without prevalent fractures would be used as the primary prevention data and the data deriving from women with prevalent fractures would be used as the secondary prevention data; if not available, we would defined relevant trials in accordance with baseline statistics as follows. Those trials in which PVF proportion of participants was equal to or more than 20% [26,27] or mean BMD T score was below -2.5 [28] when PVF proportion was available, or in which mean BMD T score was below -2.5 or mean age was above 62 years [26,27] when PVF proportion wasn't available would be defined as secondary prevention trials. On the contrary, other trials would be defined as primary prevention trials. BMD T score at the femoral neck (FN) measured by DXA was preferred [29]. BMD measured at the total hip or lumbar spine would be considered when BMD measured at the FN was unknown.

## Study selection

Two authors independently deleted duplicated records via the Endnote X9 software, and then scanned the titles and abstracts of the remaining records for the assessment of potential eligibility, and finally reviewed full-text articles for final eligibility for inclusion. Possible disagreements on study selection would be addressed through discussion between them or by another author involved if necessary.

## Data extraction and quality assessment

Two authors independently extracted the data as follows from included studies: first author name, publication year, quality assessment results, duration of drug used and RCT performed, mean age, Caucasian proportion, PVF proportion, BMD, drug name, usage and dosage of drug, sample size of each arm, and outcome data (including study-level survival data of hazard ratios (HRs) and their 95% confidence intervals (CIs) for the outcome of nonvertebral fractures, and study-level dichotomous data of total participant numbers and the numbers of participants having experienced the event of interest for the outcomes of vertebral fractures, tolerability, and acceptability).

Two independent reviewers assessed the quality of included studies based on the Jadad scale [30]. Possible total score according to this scale ranges from 0 point to 5 points. The score derives from three aspects (namely, randomization, double blinding, and withdrawals). As for randomization, a study in which an accurate randomization method was used, or in which randomization method was simply mentioned, or in which an inaccurate randomization method was used or randomization method was not mentioned, gains 2 points, or 1 point, or 0 point, respectively. Similarly, the score as for double blinding was given. As for withdrawals, a study in which the information of withdrawals and dropouts was detailedly provided, or in which not, respectively gains 1 or 0 point. Possible disagreements on data extraction or quality

assessment would be addressed through discussion between them or by another author involved if necessary.

## Statistical analysis

The Bayesian random-effects model [31] was used to perform pairwise meta-analysis and network meta-analysis stratified by the primary prevention data and the secondary prevention data to synthesize HRs and their 95% CIs for the outcome of nonvertebral fractures and to synthesize risk ratios (RRs) and their 95% CIs for the three other outcomes. 95% CI of effect size not including 1.0 means difference of statistical significance. We computed $I^2$ statistic for heterogeneity assessment, and built node-splitting model for inconsistency assessment. $I^2$ greater than 50% denotes substantial heterogeneity [32], and p value from test of inconsistency less than 0.05 denotes significant inconsistency [33]. We plotted comparison-adjusted funnel plots to assess publication bias [34].

We calculated the surface under the cumulative ranking curve (SUCRA) to help identify the best drug intervention [35]. However, the relative ranking for each of the four outcomes in this study might be different, which would make the choice of the optimal intervention challenging. Therefore, we tried to use the factor analysis method to reduce the four variables consisting of SUCRA values for four outcomes into two possible common factors. Then the Stata command of "clusterank"[34] was applied to hierarchical cluster analysis [36] using the data of the two common factors, and meanwhile a two-dimensional scatter plot was plotted to show the best treatment cluster which might contain one or greater than one drug intervention. The optimal number of clusters for this Stata command is chosen according to an internal cluster validation measure, named clustering gain [37]. This measure with a maximum value represents maximum intracluster similarity and minimum intercluster similarity. If Kaiser-Meyer-Olkin Measure of Sampling Adequacy was above 0.5, p value form Bartlett's Test of Sphericity was below 0.05, and cumulative proportion of variance explained by first two common factors was above 80% in the results of factor analysis, which suggested the appropriateness of factor analysis in this condition, the subsequent cluster analysis and the analysis of computing composite score based on common factors would be performed. Otherwise, the subsequent analysis wouldn't be performed.

We grouped included trials into the primary prevention group vs. the secondary prevention group to conduct all analyses for each outcome. Meanwhile, we only kept double-blind RCTs to conduct relevant analyses for the safety outcome of acceptability. Network meta-analyses were done using JAGS (version 4.3.0) and R (version 3.6.0), factor analyses were done using SPSS (version 24.0), and network plots, clustered ranking plots and comparison-adjusted funnel plots were plotted using Stata (version 15.1).

## Results

### Summary of included studies

A total of 367 full-text articles were reviewed for eligibility, of which 57 were included for quantitative synthesis (Fig 1). The included 57 RCTs are listed in S3 Appendix. Baseline characteristics, outcome data and quality assessment results of these studies are given in S4 Appendix. Included studies had an average Jadad score of 4.2 points (standard deviation: 1.1), and the score of most included studies (54/57) was equal to or greater than 3 points, which indicated that primary studies included in this study had high quality in general. The average duration of drug interventions used for participants was 27.5 months (standard deviation: 16.3), and the duration was in the range of 12 to 72 months. Of included studies, 19 provided the

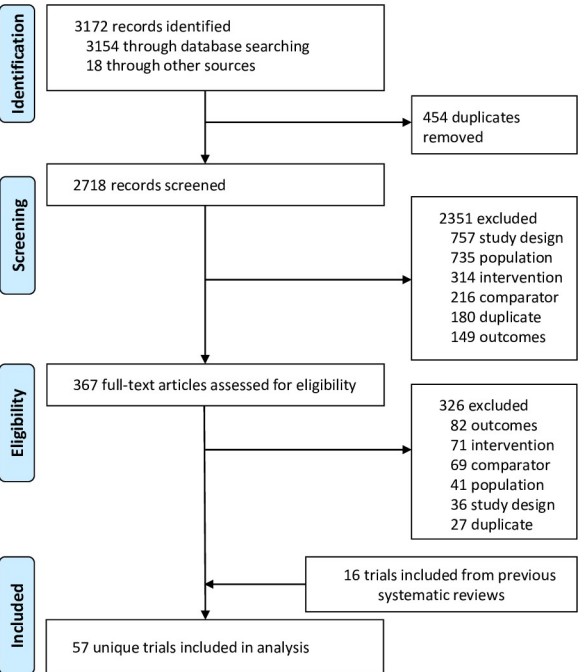

**Fig 1. Flow chart of study selection.**

primary prevention data, 42 provided the secondary prevention data, and 4 provided both of the two kinds of data.

In included RCTs, mean age had an average value of 67.5 years (standard deviation: 5.5) in the range of 53.5 to 85.5 years; and the proportion of Caucasians had an average value of 90.9% (standard deviation: 9.9%) in the range of 60.4% to 100%. The four outcomes of interest in this study totally involved 16 drug interventions (i.e., 15 active interventions and an inactive intervention of placebo) and 106320 PMW. The primary prevention trials in total involved 29642 PMW and 10 interventions (i.e., alendronate, bazedoxifene, denosumab, ibandronate, PTH (1–84), raloxifene, risedronate, zoledronate (once per 18 months), zoledronate (once per 24 months), and placebo), while the secondary prevention trials in total involved 76678 PMW and 15 interventions (i.e., abaloparatide, alendronate, bazedoxifene, denosumab, ibandronate, lasofoxifene, PTH (1–84), raloxifene, risedronate, romosozumab, strontium ranelate, teriparatide, zoledronate (once per 12 months), zoledronate (once per 24 months), and placebo).

Fig 2A–2H are network plots of vertebral fractures (A), nonvertebral fractures (B), tolerability (C) and acceptability (D) in the primary prevention group, and vertebral fractures (E), nonvertebral fractures (F), tolerability (G) and acceptability (H) in the secondary prevention group. There is no closed loop in Fig 2D, whereas there are at least two closed loops in any other network plot. It meant that test of inconsistency was necessarily conducted for all outcomes in the two subgroups except for acceptability in the primary prevention group. In addition, placebo was most commonly compared to other interventions in each of these network plots. On the contrary, the lines among active drugs were thin, and the number of those lines was limited, which suggested the lack of head-to-head trials comparing active drugs.

## Subgroup network meta-analyses

Fig 3 (active drugs compared to placebo) and S5–S8 Appendixes (active drugs compared among them) show the results of network meta-analysis based on the primary prevention data.

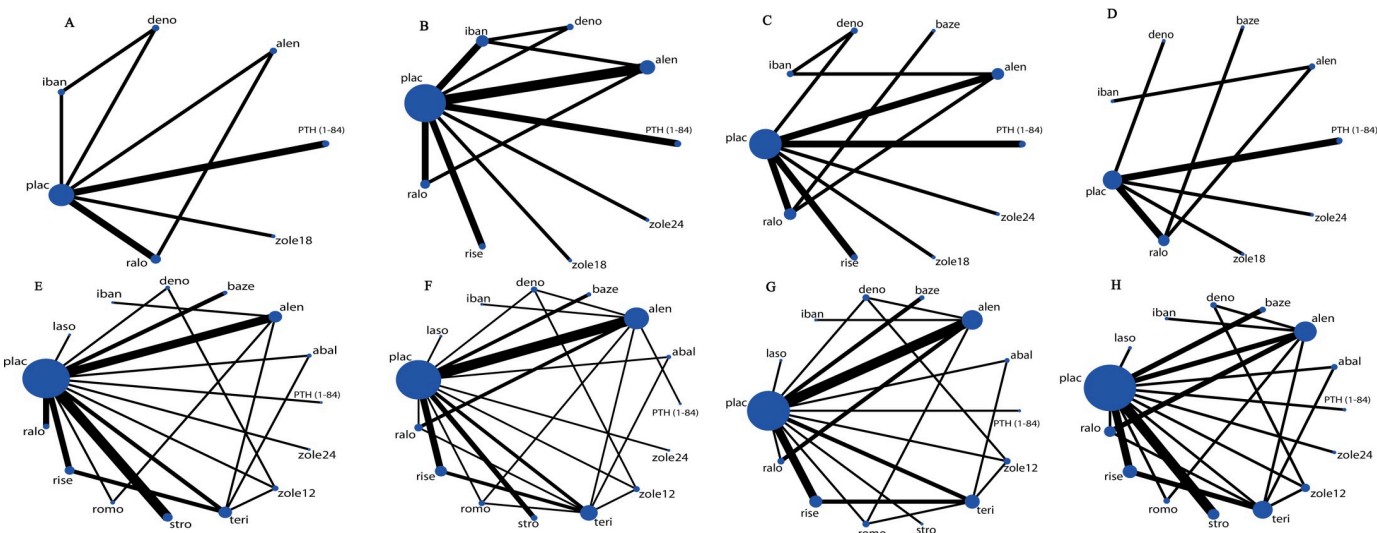

**Fig 2.** Network plots of vertebral fractures (A), nonvertebral fractures (B), tolerability (C) and acceptability (D) in the primary prevention group, and vertebral fractures (E), nonvertebral fractures (F), tolerability (G) and acceptability (H) in the secondary prevention group. PTH (1–84), parathyroid hormone (1–84); romo, romosozumab; abal, abaloparatide; teri, teriparatide; zole24, zoledronate (5 mg per 24 months, intravenous); zole18, zoledronate (5 mg per 18 months, intravenous); zole12, zoledronate (5 mg per 12 months, intravenous); deno, denosumab; stro, strontium ranelate; laso, lasofoxifene; baze, bazedoxifene; ralo, raloxifene; plac, placebo; alen, alendronate; iban, ibandronate; rise, risedronate. The size of every circle is proportional to the number of randomly assigned participants and indicates the sample size; the width of the lines corresponds to the number of trials.

In terms of vertebral fractures (Fig 3A and S5 Appendix), alendronate (RR 0.60, 95% CI 0.42–0.84), PTH (1–84) (RR 0.27, 95% CI 0.11–0.61), raloxifene (RR 0.57, 95% CI 0.47–0.73), and zoledronate (once per 18 months) (RR 0.46, 95% CI 0.28–0.74) reduced fracture risk compared with placebo. Meanwhile, no significant difference was found in comparisons among active drugs. In terms of nonvertebral fractures (Fig 3B and S6 Appendix), zoledronate (once per 18 months) reduced fracture risk compared with placebo (HR 0.66, 95% CI 0.51–0.85) or raloxifene (HR 0.71, 95% CI 0.54–0.94). In terms of tolerability (Fig 3C and S7 Appendix) and acceptability (Fig 3D and S8 Appendix), no significant difference was observed in all comparisons.

Fig 4 (active drugs compared to placebo) and S9–S12 Appendixes(active drugs compared among them) show the results of network meta-analysis based on the secondary prevention data. In terms of vertebral fractures (Fig 4A and S9 Appendix), abaloparatide (RR 0.17, 95% CI 0.049–0.40), alendronate (RR 0.62, 95% CI 0.51–0.78), bazedoxifene (RR 0.62, 95% CI 0.46–0.83), denosumab (RR 0.32, 95% CI 0.23–0.43), lasofoxifene (RR 0.60, 95% CI 0.45–0.79), PTH (1–84) (RR 0.44, 95% CI 0.20–0.95), raloxifene (RR 0.69, 95% CI 0.53–0.90), risedronate (RR 0.59, 95% CI 0.47–0.74), romosozumab (RR 0.30, 95% CI 0.23–0.41), strontium ranelate (RR 0.72, 95% CI 0.63–0.81), teriparatide (RR 0.26, 95% CI 0.19–0.35), and zoledronate (once per 12 months) (RR 0.31, 95% CI 0.24–0.44) reduced fracture risk compared with placebo. Meanwhile, abaloparatide, denosumab, romosozumab, teriparatide, and zoledronate (once per 12 months) were superior to alendronate, bazedoxifene, lasofoxifene, raloxifene, risedronate, and strontium ranelate; and zoledronate (once per 24 months) was inferior to abaloparatide. In terms of nonvertebral fractures (Fig 4B and S10 Appendix), abaloparatide (HR 0.54, 95% CI 0.31–0.96), alendronate (HR 0.81, 95% CI 0.70–0.94), denosumab (HR 0.80, 95% CI 0.64–0.99), lasofoxifene (HR 0.76, 95% CI 0.60–0.96), risedronate (HR 0.70, 95% CI 0.57–0.86), romosozumab (HR 0.69, 95% CI 0.56–0.84), teriparatide (HR 0.63, 95% CI 0.48–0.81), and zoledronate (once per 12 months) (HR 0.76, 95% CI 0.63–0.94) reduced fracture risk

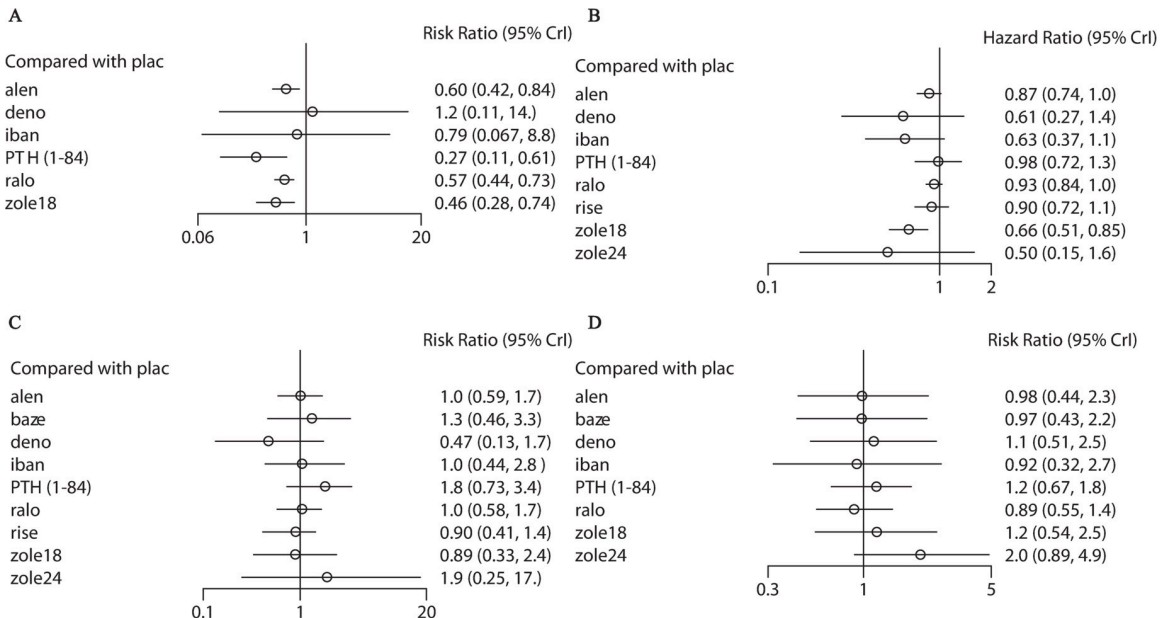

**Fig 3.** Network meta-analysis for vertebral fractures (A), nonvertebral fractures (B), tolerability (C) and acceptability (D) in the primary prevention group with active drugs compared to placebo. PTH (1–84), parathyroid hormone (1–84); zole24, zoledronate (5 mg per 24 months, intravenous); zole18, zoledronate (5 mg per 18 months, intravenous); deno, denosumab; baze, bazedoxifene; ralo, raloxifene; plac, placebo; alen, alendronate; iban, ibandronate; rise, risedronate.

compared with placebo. Meanwhile, risedronate, romosozumab and teriparatide were superior to bazedoxifene and raloxifene; and strontium ranelate was inferior to teriparatide. In terms of tolerability (Fig 4C and S11 Appendix), abaloparatide (RR 1.7, 95% CI 1.2–2.3) and PTH (1–84) (RR 1.9, 95% CI 1.4–2.7) increased the risk of withdrawals due to adverse events compared with placebo. Meanwhile, abaloparatide and PTH (1–84) were less tolerated than alendronate, bazedoxifene, lasofoxifene, risedronate, romosozumab, and teriparatide; and PTH (1–84) was less tolerated than denosumab, ibandronate, raloxifene, strontium ranelate, and zoledronate (once per 12 months) as well. In terms of acceptability (Fig 4D and S12 Appendix), PTH (1–84) (RR 1.2, 95% CI 1.1–1.4) increased the risk of withdrawals due to any cause compared with placebo, whereas risedronate (RR 0.87, 95% CI 0.79–0.95) reduced this risk. Meanwhile, abaloparatide and PTH (1–84) were less acceptable than alendronate, bazedoxifene, raloxifene, risedronate, strontium ranelate, and teriparatide; and risedronate was more acceptable than bazedoxifene and lasofoxifene.

## Assessment of inconsistency and heterogeneity

S13–S15 Appendixes provide the results from test of inconsistency for vertebral fractures (S13 Appendix), nonvertebral fractures (S14 Appendix), and tolerability (S15 Appendix) in the primary prevention group. The p values from test of inconsistency in this subgroup ranged from 0.244 to 0.993. Test of inconsistency for acceptability in this subgroup wasn't performed due to the lack of evidence loop in this outcome network. S16–S19 Appendixes provide the results from test of inconsistency for vertebral fractures (S16 Appendix), nonvertebral fractures (S17 Appendix), tolerability (S18 Appendix), and acceptability (S19 Appendix) in the secondary prevention group. The p values from test of inconsistency in this subgroup ranged from 0.051 to 0.999. In summary, all p values from test for inconsistency in this study were greater than

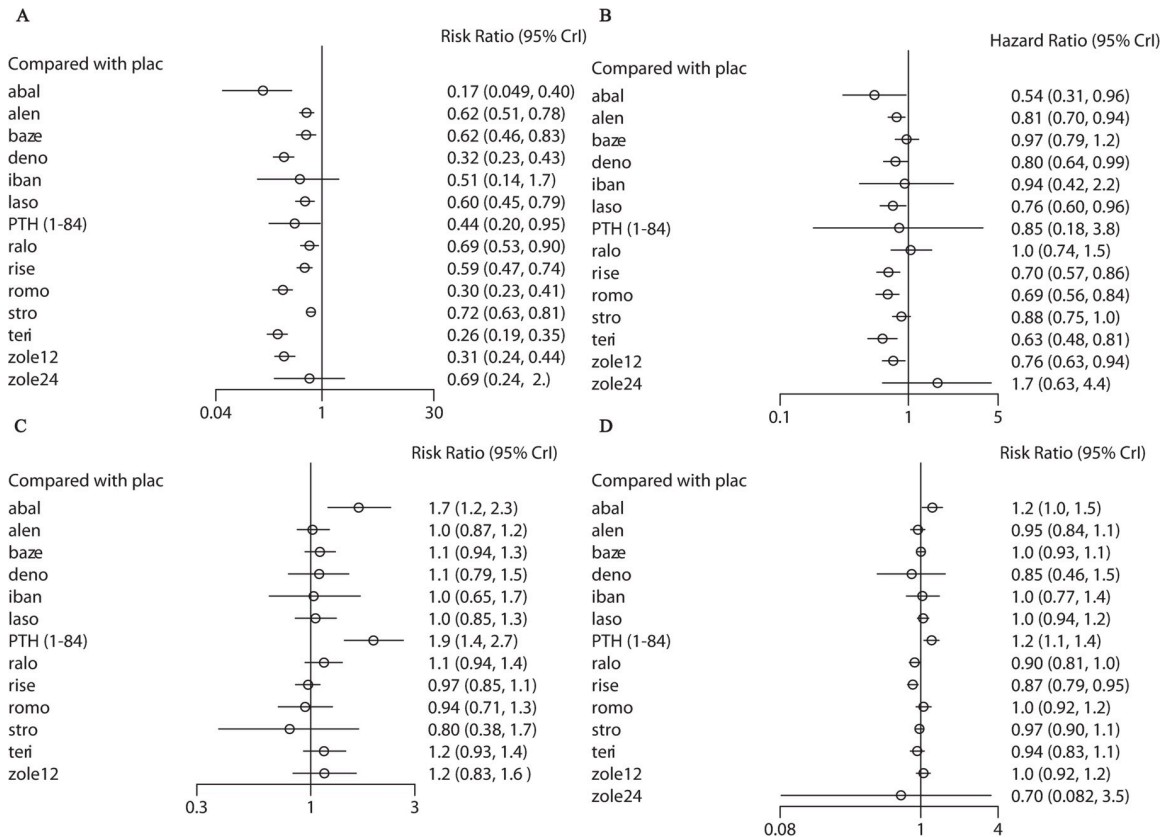

**Fig 4.** Network meta-analysis for vertebral fractures (A), nonvertebral fractures (B), tolerability (C) and acceptability (D) in the secondary prevention group with active drugs compared to placebo. PTH (1–84), parathyroid hormone (1–84); romo, romosozumab; abal, abaloparatide; teri, teriparatide; zole24, zoledronate (5 mg per 24 months, intravenous); zole12, zoledronate (5 mg per 12 months, intravenous); deno, denosumab; stro, strontium ranelate; laso, lasofoxifene; baze, bazedoxifene; ralo, raloxifene; plac, placebo; alen, alendronate; iban, ibandronate; rise, risedronate.

0.05, which suggested there was no significant inconsistency in network meta-analysis for any outcome in the two different subgroups.

S20–S27 Appendixes provide the results of heterogeneity assessment in this network meta-analysis. No substantial heterogeneity was observed in network meta-analysis for vertebral fractures (S20 Appendix), nonvertebral fractures (S21 Appendix) and acceptability (S23 Appendix) in the primary prevention group, and for nonvertebral fractures (S25 Appendix) and tolerability (S26 Appendix) in the secondary prevention group according to the $I^2$ statistic <50%.

Although there was substantial heterogeneity ($I^2$ = 81.1%) for risedronate vs. placebo in pairwise meta-analysis of tolerability in the primary prevention group (S22 Appendix), the estimate value of RR in any primary study was not statistically significant, which was essentially consistent with the result of pooled direct estimate or network estimate as for this comparison. Although there was substantial heterogeneity ($I^2$ = 87.4%) for zoledronate (once per 12 months) vs. placebo in network meta-analysis of vertebral fractures in the secondary prevention group (S24 Appendix), the indirect estimate with wide 95% CI (representing the lack of statistical power) was invalid and the results of pooled direct estimate and network estimate were similar as for this comparison. Although there was substantial heterogeneity ($I^2$ = 55.8%) for teriparatide vs. raloxifene in network meta-analysis of acceptability in the secondary

prevention group (S27 Appendix), the estimate value of RR resulting from either direct evidence or indirect evidence was not statistically significant, which was essentially consistent with the result of pooled network estimate as for this comparison. Therefore, these results of network estimate deriving from the three different comparisons were still valid despite the existence of substantial heterogeneity in these comparisons.

## Factor analyses and cluster analysis

S28 Appendix presents SUCRA values of drug interventions for 4 different outcomes estimated based on the primary prevention data. The results (S29 Appendix) of factor analysis on these SUCRA values show that Kaiser-Meyer-Olkin Measure of Sampling Adequacy was equal to 0.470 ($<$0.5) and p value form Bartlett's Test of Sphericity was equal to 0.797 ($>$0.05), which suggested the data were not suitable for factor analysis. Thus, common factors extracted in this factor analysis were not used to perform cluster analysis.

S30 Appendix presents SUCRA values of drug interventions for 4 different outcomes estimated based on the secondary prevention data. The results (S31 Appendix) of factor analysis on these SUCRA values show that Kaiser-Meyer-Olkin Measure of Sampling Adequacy was equal to 0.509 ($>$0.5) and p value form Bartlett's Test of Sphericity was equal to 0.002 ($<$0.05), which suggested the data were suitable for factor analysis. Meanwhile, Factor 1 and Factor 2 totally explained 84.67% of the total variance deriving from 4 primary variables, which implied the appropriateness of the two common factors, as substitutes for 4 primary variables, used for cluster analysis. The rotated component matrix in S31 Appendix shows the correlation coefficient between Factor 1 and SUCRA for vertebral fractures or for nonvertebral fractures approximated to 1, which indicated that Factor 1 represented the anti-fracture efficacy of drug interventions. Similarly, Factor 2 represented safety. The data of Factor 1 (efficacy score) and Factor 2 (safety score) are provided in S30 Appendix. The safety score of risedronate was greater than its efficacy score, whereas the efficacy scores of romosozumab, teriparatide, denosumab, abaloparatide, and zoledronate (once per 12 months) were greater than their safety scores. The result of cluster analysis based on the two common factors is shown in Fig 5. All drug interventions were optimally clustered into 6 groups since in this condition the statistic of clustering gain had the maximum value of 14.19. The Cluster 1 consisting of romosozumab, teriparatide, denosumab, and risedronate simultaneously had superior efficacy and safety. The Cluster 2 consisting of abaloparatide and zoledronate (once per 12 months) had the similar efficacy and the inferior safety compared to Cluster 1. The four other clusters had the inferior effectiveness compared to Cluster 1 or Cluster 2, and had the non-superior safety compared to Cluster 1 in general. Additionally, composite scores computed based on Factor 1 and Factor 2 are listed in S30 Appendix, and the four drugs in Cluster 1 had the greatest composite scores.

## Detection of publication bias

Comparison-adjusted funnel plots are provided in S32 Appendix, and all of them did not suggest dominant publication bias.

## Discussion

### Summary of evidence

This study is the first one in which evidence of the comparative efficacy and safety of 16 drug interventions (15 active interventions and placebo) was respectively provided for primary and secondary prevention of PMO via network meta-analysis, and this study is also the first one in which the optimal drug interventions in the condition of efficacy and safety simultaneously

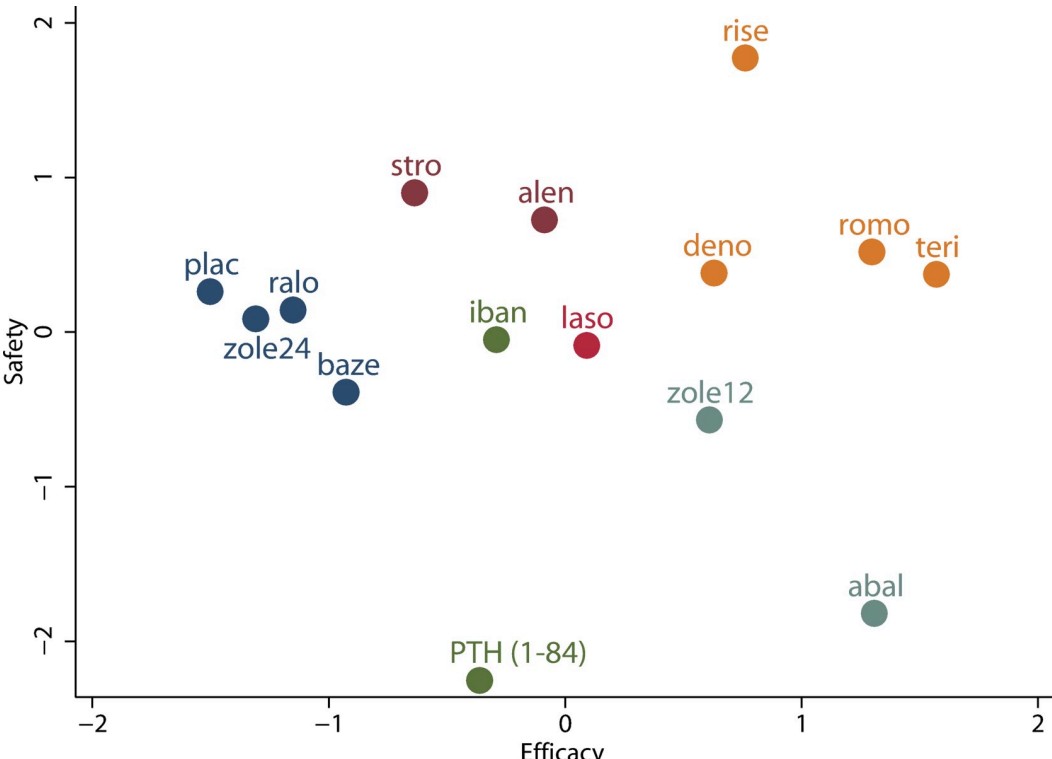

**Fig 5. Clustered ranking plot of drug interventions for secondary prevention of osteoporotic fractures.** PTH (1–84), parathyroid hormone (1–84); romo, romosozumab; abal, abaloparatide; teri, teriparatide; zole24, zoledronate (5 mg per 24 months, intravenous); zole12, zoledronate (5 mg per 12 months, intravenous); deno, denosumab; stro, strontium ranelate; laso, lasofoxifene; baze, bazedoxifene; ralo, raloxifene; plac, placebo; alen, alendronate; iban, ibandronate; rise, risedronate.

taken into account were identified for secondary prevention of PMO via factor analysis and cluster analysis. In summary, there are the following main findings in this study.

For secondary prevention of PMO, the cluster analysis results showed that romosozumab, teriparatide, denosumab and risedronate constituted Cluster 1 which simultaneously had superior efficacy and safety, and meanwhile the same four drug interventions had the greatest composite scores that were computed based on the results of factor analysis. It means that these four drug interventions are the optimal options for secondary prevention of osteoporotic fractures among PMW in the context of both efficacy and safety considered. It is worth mentioning that in this study the safety outcomes consisted of tolerability and acceptability, and failed to consider the severity of adverse effects. Furthermore, the factor analysis results revealed that the safety score of risedronate was greater than its efficacy score and there was a contrary case with the three other treatments. Hence, risedronate was able to enter into the optimal cluster with superior efficacy and safety, which mainly depended on its advantage in safety; whereas the other three interventions mainly depended on their advantage in efficacy. This point is also supported by the following results found by network meta-analysis in our study. Compared with placebo, romosozumab, teriparatide, and denosumab had the similar tolerability and acceptability, and risedronate had the similar tolerability and superior acceptability, which supports risedronate with advantage in safety. Although risedronate reduced the risk of vertebral fractures (RR 0.59) and nonvertebral fractures (HR 0.70) compared with placebo, romosozumab (HR 0.69) and teriparatide (HR 0.63) had greater risk reduction as for nonvertebral fractures while romosozumab (RR 0.30), teriparatide (RR 0.26) and denosumab

(RR 0.32) had greater risk reduction as for vertebral fractures, which supports these three drug treatments with advantage in efficacy. These effect sizes of the four treatments for reducing vertebral and nonvertebral fractures are similar with those estimated in two prior studies [14,15] in which network meta-analyses were performed based on the combined data for primary and secondary prevention of osteoporotic fractures in PMW.

For secondary prevention of PMO, oral risedronate was observed to have superiority over the other oral bisphosphonates (i.e., alendronate, and ibandronate) in efficacy and safety in the clustered ranking plot, which was mainly because compared with alendronate or ibandronate risedronate had greater risk reduction in terms of nonvertebral fractures (vs. placebo, HRs for risedronate, alendronate and ibandronate: 0.70, 0.81 and 0.94) and withdrawals resulting from adverse events (vs. placebo, RRs for risedronate, alendronate and ibandronate: 0.97, 1.0 and 1.0) or any cause (vs. placebo, RRs for risedronate, alendronate and ibandronate: 0.87, 0.95 and 1.0) despite these differences without statistical significance. Similarly, risedronate with statistically insignificant advantage over alendronate and ibandronate in reducing nonvertebral fractures was demonstrated in several previous network meta-analyses which enrolled patients with PMO [14,19,38] or with osteoporosis of any cause [39,40], and in reducing adverse events was demonstrated in two other studies [18,41] which enrolled patients with primary osteoporosis.

For secondary prevention of PMO, abaloparatide and zoledronate (once per 12 months) constituted Cluster 2 which had the similar efficacy and the inferior safety compared to Cluster 1. Their superior efficacy derived from the ability of significantly reducing vertebral fractures (vs. placebo, RRs for abaloparatide and zoledronate: 0.17 and 0.31) and nonvertebral fractures (vs. placebo, HRs for abaloparatide and zoledronate: 0.54 and 0.76). These effect size estimates of abaloparatide and zoledronate approximate to those calculated in three previous studies [14,15,42]. Meanwhile, they significantly or insignificantly increased withdrawals deriving from adverse events (vs. placebo, abaloparatide: RR 1.7, 95% CI 1.2–2.3); zoledronate: RR 1.2, 95% CI 0.83–1.6)), which led to their inferior safety. Consistent with the results, abaloparatide was found to increase this withdrawal risk (vs. placebo: RR 1.62, 95% CI 1.15–2.27; vs. teriparatide: RR 1.44, 95% CI 1.04–2.00) in a primary study [43], while zoledronate was found to, with statistical significance, increase the risk of adverse events in two secondary studies [16,18].

For primary prevention of PMO, oral bisphosphonates (i.e., alendronate, risedronate and ibandronate) were not observed to reduce vertebral and nonvertebral fractures compared to placebo except that alendronate reduced vertebral fractures (RR 0.60, 95% CI 0.42–0.84) in our study. Similar results were found in two Cochrane systematic reviews [26,27] in which in terms of primary prevention of osteoporotic fractures in PMW alendronate showed a significant reduction in vertebral fractures (RR 0.55, 95% CI 0.38–0.80) but not in nonvertebral fractures (RR 0.89, 95% CI 0.76–1.04) [26] while risedronate didn't show a significant reduction in vertebral (RR 0.97, 95% CI 0.42–2.25) and nonvertebral fractures (RR 0.81, 95% CI 0.25–2.58) [27]. In the primary prevention group of this study, zoledronate (once per 18 months) was the only drug intervention which reduced the risk of both vertebral fractures (RR 0.46, 95% CI 0.28–0.74) and nonvertebral fractures (HR 0.66, 95% CI 0.51–0.85), and meanwhile didn't increase withdrawal risk. This evidence on zoledronate used at a dose of 5 mg at 18-month intervals derives from a recently-published well-designed RCT [22] which mainly focused on PMW with osteopenia and without PVF, and in which compared to placebo zoledronate led to an increase in BMD and a decrease in markers of bone turnover, apart from having lowered fracture risk.

## Clinical implications

For secondary prevention of osteoporotic fractures in PMW, when both anti-fracture efficacy and safety is taken into consideration our study supports that the best drug treatments are

romosozumab, teriparatide, denosumab and risedronate, of which risedronate has greater safety superiority, and the others have greater efficacy superiority. It implies that risedronate is the optimal drug intervention for PMW with mild to moderate osteoporosis, while the three others are the optimal drug interventions for PMW with severe osteoporosis or at high risk for developing fragility fractures. In clinical reality, anabolic agents are preferred in severe osteoporosis patients due to their capability of reducing fracture risk more quickly and to a greater extent than antiresorptive agents [44]. However, fractures and bone loss may rapidly occur after bone-building drugs such as teriparatide are discontinued [45,46]. For preventing this reversal effect, a sequential treatment strategy, starting with a bone-forming agent, followed by an antiresorptive, is recommended for those patients [1,47]. Nowadays, there are five sequential treatments by which antiresorptives are able to maintain or further increase BMD gains obtained by anabolic drugs, and they are respectively 1 year of alendronate switching from 1 year of PTH (1–84) [48], 2 years of denosumab switching from 2 years of teriparatide [49], 2 years of alendronate switching from 1 year of romosozumab [50], 1 year of denosumab switching from 1 year of romosozumab [51], and 24 months of alendronate switching from18 months of abaloparatide [52]. Which sequential treatment is the best remains unclear, and the findings in our study can help address this issue. The two treatment strategies starting with PTH (1–84) or abaloparatide shouldn't be considered at first owing to the poor safety and anti-fracture efficacy with PTH (1–84) and the poor safety with abaloparatide found in our study. On the contrary, the treatment strategies starting with teriparatide or romosozumab should be preferred owing to their superior anti-fracture efficacy and safety found in our study. Based on RCTs, evidence about teriparatide followed by denosumab with significant risk reduction in fractures is lacking [49], whereas there is clear evidence that the sequential treatment starting with romosozumab then switching to alendronate or denosumab has significantly reduced fracture risk [50,51]. Besides, the combination of romosozumab and bisphosphonates (e.g., alendronate) might reduce the risk of cardiovascular adverse events [53]. Accordingly, the sequential treatment starting with romosozumab followed by alendronate could be the best one for PMW with severe osteoporosis or at high risk for osteoporotic fractures.

For secondary prevention of mild to moderate osteoporosis, oral bisphosphonates are often recommended as first-line agents. There are some osteoporotic patients for whom oral bisphosphonates are not enough effective, such as, fragility fractures occurring during the treatment, and BMD having an unsatisfactory response. Among this population denosumab was demonstrated to lead to greater increase in BMD than bisphosphonates did in Lyu et al.'s study [54], in which, however, evidence on the comparative anti-fracture effectiveness was limited. Our study demonstrated that a significantly greater risk reduction in vertebral fractures derived from denosumab than derived from oral bisphosphonates (vs. alendronate, RR 0.51; vs. risedronate, RR 0.53), which, to some degree, further supports denosumab used in osteoporotic PMW on which a bisphosphonate hasn't resulted in a satisfactory effect. Similar with teriparatide, there are reverse effects on bone markers, BMD and fracture risk after denosumab discontinued [46,55]. Thus, other agents, mainly bisphosphonates, should be immediately given in this case.

For primary prevention of osteoporotic fractures in PMW, oral bisphosphonates are widely used in clinical practice. However, there hasn't been evidence deriving from RCTs that a bisphosphonate could significantly reduce both vertebral and nonvertebral fractures until now. Fortunately, we have identified the newest evidence on zoledronate intravenously used at a dose of 5 mg per 18 months, which is the only drug intervention which has been shown to significantly reduce both vertebral (RR 0.46) and nonvertebral fracture risk (HR 0.66) for primary

prevention of PMO, and which will help improve the management of PMO in terms of primary prevention due to its superior anti-fracture efficacy and good safety profile.

For primary or secondary prevention of osteoporotic fractures in PMW, although the evidence of vitamin D and calcium used for the treatment of fractures remains limited their supplements should be considered, given that we included those trials in which all women received calcium and/or vitamin D, and that the supplements are recommended for adults with no known osteoporosis or vitamin D deficiency to prevent fractures [56]. Meanwhile, other important points (e.g., drug compliance, administration route, cost effectiveness, patient comorbidities, and a clinician's familiarity with use of a particular treatment) should be also considered in a particular clinical setting, apart from anti-fracture efficacy and safety assessed in our study. Moreover, we included those RCTs which mainly enrolled Caucasian women, which implies evidence on the comparative efficacy and safety of osteoporosis drugs identified in this study is most applicable to white PMW.

## Strengths and limitations

The greatest strength of our study is that we assessed the comparative efficacy and safety of different agent interventions for PMO respectively in the primary prevention group and in the secondary prevention group, and identified the optimal treatments for secondary prevention in consideration of both effectiveness and safety via factor and cluster analysis on the data of SUCRA values deriving from network meta-analysis. Meanwhile, we synthesized the up-to-date evidence deriving from newly-published RCTs [22,43,50,51,57] mainly involving two newly approved drugs (namely, romosozumab [23] and abaloparatide [24]) and a new drug intervention (namely, zoledronate 5 mg per 18 months), which could become a strong basis of updating the 2017 ACP guideline [13]. Second, no significant inconsistency existed in all network meta-analyses in this study, and no substantial heterogeneity existed in the majority of network meta-analyses while in the minority of them existed the heterogeneity that didn't affect the validity of network estimates. Meanwhile, RCTs included in our study generally had high quality, and no dominant publication bias was found. These facts resulted in the validity of this network meta-analysis. Third, we improved the accuracy of network meta-analysis results by doing the following things: ① an identical drug used at different intervals was taken for different interventions, such as zoledronate (once per 12 months), zoledronate (once per 18 months) and zoledronate (once per 24 months) were considered as three interventions in this study, although alendronate used at a dose of 5 or 10 mg was taken for one intervention because the anti-fracture efficacy of these two dosages was combined in included trials [58,59]; ② we only kept double-blind RCTs for the assessment of comparative acceptability; ③ we used the fixed-effects model to perform network meta-analyses for vertebral fractures and nonvertebral fractures in the primary prevention group, given the presence of only low or no inconsistency and heterogeneity in those analyses; and ④ we synthesized HRs (instead of RRs) for nonvertebral fractures since primary studies had provided study-level survival data for this outcome and survival data contains more information (i.e., the time when an event of interest occurs) than dichotomous data does. Fourth, we further improved the network transitivity in this study compared to prior network meta-analyses [14,15,42] by only considering those trials which mainly enrolled white people given that the majority of participants were Caucasians in those trials of new drug interventions (e.g., zoledronate used once per 18 months [22], romosozumab [50,51], and abaloparatide [43]), by only considering those trials in which all participants were women with primary osteoporosis since those new drug trials [22,43,50,51] completely enrolled this type of patients, and by grouping all trials into the primary prevention subgroup and the secondary prevention subgroup to increase the comparability of baseline characteristics inside each subgroup.

This study has some limitations. First, for primary prevention of osteoporotic fractures, we failed to find evidence about the efficacy of three bone-building drugs (i.e., abaloparatide, teriparatide and romosozumab), and failed to observe denosumab with statistically significant efficacy since its wide 95% CIs across 1.0 suggested the lack of statistical power. Given that these four drugs were observed to have superior effectiveness for secondary prevention of osteoporotic fractures, future studies with adequate statistical power are needed to clarify whether it is possible to use them for primary prevention by lowering their dosages or by prolonging their dose intervals, such as zoledronate used once per 12 months is effective for secondary prevention while this drug used once per 18 months is effective for primary prevention. Second, we failed to perform factor and cluster analysis to synthesize evidence of efficacy and safety to help identify the optimal interventions for primary prevention owing to the data limited. Instead, we identified that the only drug intervention of zoledronate (once per 18 months) reduced the risk of both vertebral and nonvertebral fractures and had the similar safety as placebo. Third, calcitonin and hormone replacement therapy were not considered in this study owing to the fact that the former is not widely used for treatment of osteoporosis any longer [13] and the latter has a limited indication for osteoporosis treatment [60,61]. Although strontium ranelate is not any more in clinical use in most countries, we still assessed it in this study since this drug was also considered in at least one of recently-published network meta-analyses [14,15,42]. Other drugs [i.e., lasofoxifene and PTH (1–84)] not approved by the U.S. Food and Drug Administration (FDA) were assessed in this study for the same reason. Fourth, although the cluster analysis in the paper ensured maximum intracluster similarity and minimum intercluster similarity according to the maximum of clustering gain, it failed to tell us whether the difference among various clusters was statistically significant or not. Thus, the statistical significance of difference among active drugs needs to be identified by more head-to-head trials. Fifth, we used tolerability and acceptability as safety endpoints, but failed to perform analysis on any of specific adverse events which we should be cautious of in clinical reality, such as romosozumab causing cerebrovascular and cardiovascular events [62], abaloparatide and teriparatide causing hypercalcemia [24], denosumab or bisphosphonates causing atypical femoral fractures (AFF) and osteonecrosis of the jaw (ONJ) [55], and bisphosphonates causing gastrointestinal side effects [55]. The two safety outcomes failed to consider the severity of adverse effects. Thus, it is meaningful that future studies assess specific adverse events and serious adverse effects. Sixth, this study failed to compare different sequential treatments owing to the absence of head-to-head trials assessing their anti-fracture efficacy. Given that an anabolic or an antiresorptive agent has the limited ability to reduce fracture risk and has the limited usage time (e.g., anabolics restricted to 2 years, and bisphosphonates restricted to 3 to 5 years) [1], and that the discontinuation of some osteoporosis drugs (e.g., teriparatide, and denosumab) may immediately cause bone loss and fracture occurrence [45,46,55], appropriate sequential treatments are considered to be essential for the long-term management of PMO [1,44,47]. Therefore, further studies are needed on which one is the optimal sequential treatment and how is the optimal timing and dosing of anti-osteoporotic agents used as sequential therapy to maintain the maximum decrease in fracture risk. Seventh, since there was a limited number of those papers with subgroup analyses done stratified by women with or without prevalent fractures, a minority of PMW grouped into the primary prevention group were women with prevalent fractures and a minority of PMW grouped into the secondary prevention group were women without prevalent fractures. Thus, the relative efficacy of drugs severally for primary prevention and secondary prevention identified in this study only approximates to the real situation, and accordingly, it requires to be validated by analyzing individual patient data. Eighth, further studies should use evidence at different time to stratify to compare different anti-osteoporotic interventions in order to provide more accurate estimates for differences between

short-term interventions (e.g., romosozumab, abaloparatide and teriparatide) and chronic interventions (e.g., oral bisphosphonates and denosumab). Ninth, we failed to consider including hip fractures as an endpoint of interest; thus, it is meaningful that further studies compare anti-osteoporosis drugs in preventing hip fractures. Last, SUCRA does not consider the magnitude of differences in effects between treatments, and fails to tell us whether the differences in the rankings based on SUCRA are with statistical significance.

## Conclusions

Based on current evidences, zoledronate intravenously used at a dose of 5 mg per 18 months, with the similar safety as placebo, is the only drug intervention which has been shown to significantly reduce both vertebral and nonvertebral fracture risk for primary prevention of osteoporotic fractures in PMW; while romosozumab, teriparatide, denosumab, and risedronate are the optimal treatments for secondary prevention when anti-fracture efficacy and safety both considered. Romosozumab followed by alendronate could be the best sequential treatment for PMW with severe osteoporosis or at high risk for osteoporotic fractures. Future studies are needed to clarify whether it is possible to use bone-building drugs for primary prevention by lowering their dosages or prolonging their dose intervals, and to identify the optimal sequential treatment and the optimal timing and dosing of sequential treatments.

## Supporting information

**S1 Appendix. PRISMA checklist.**
(DOC)

**S2 Appendix. Full search strategy.**
(PDF)

**S3 Appendix. Checklist of studies included for network meta-analysis.**
(PDF)

**S4 Appendix. Study characteristics, outcome data and quality assessment.**
(XLSX)

**S5 Appendix. Network meta-analysis for vertebral fractures in the primary prevention group with active drugs compared among them.**
(PDF)

**S6 Appendix. Network meta-analysis for nonvertebral fractures in the primary prevention group with active drugs compared among them.**
(PDF)

**S7 Appendix. Network meta-analysis for tolerability in the primary prevention group with active drugs compared among them.**
(PDF)

**S8 Appendix. Network meta-analysis for acceptability in the primary prevention group with active drugs compared among them.**
(PDF)

**S9 Appendix. Network meta-analysis for vertebral fractures in the secondary prevention group with active drugs compared among them.**
(PDF)

**S10 Appendix. Network meta-analysis for nonvertebral fractures in the secondary prevention group with active drugs compared among them.**
(PDF)

**S11 Appendix. Network meta-analysis for tolerability in the secondary prevention group with active drugs compared among them.**
(PDF)

**S12 Appendix. Network meta-analysis for acceptability in the secondary prevention group with active drugs compared among them.**
(PDF)

**S13 Appendix. Inconsistency test for vertebral fractures in the primary prevention group.**
(PDF)

**S14 Appendix. Inconsistency test for nonvertebral fractures in the primary prevention group.**
(PDF)

**S15 Appendix. Inconsistency test for tolerability in the primary prevention group.**
(PDF)

**S16 Appendix. Inconsistency test for vertebral fractures in the secondary prevention group.**
(PDF)

**S17 Appendix. Inconsistency test for nonvertebral fractures in the secondary prevention group.**
(PDF)

**S18 Appendix. Inconsistency test for tolerability in the secondary prevention group.**
(PDF)

**S19 Appendix. Inconsistency test for acceptability in the secondary prevention group.**
(PDF)

**S20 Appendix. Heterogeneity assessment for vertebral fractures in the primary prevention group.**
(PDF)

**S21 Appendix. Heterogeneity assessment for nonvertebral fractures in the primary prevention group.**
(PDF)

**S22 Appendix. Heterogeneity assessment for tolerability in the primary prevention group.**
(PDF)

**S23 Appendix. Heterogeneity assessment for acceptability in the primary prevention group.**
(PDF)

**S24 Appendix. Heterogeneity assessment for vertebral fractures in the secondary prevention group.**
(PDF)

**S25 Appendix. Heterogeneity assessment for nonvertebral fractures in the secondary prevention group.**
(PDF)

**S26 Appendix. Heterogeneity assessment for tolerability in the secondary prevention group.**
(PDF)

**S27 Appendix. Heterogeneity assessment for acceptability in the secondary prevention group.**
(PDF)

**S28 Appendix. SUCRA values of drug interventions for primary prevention of osteoporotic fractures.**
(XLSX)

**S29 Appendix. Factor analysis of SUCRA values estimated based on the primary prevention data.**
(PDF)

**S30 Appendix. SUCRA values of drug interventions for secondary prevention of osteoporotic fractures.**
(XLSX)

**S31 Appendix. Factor analysis of SUCRA values estimated based on the secondary prevention data.**
(PDF)

**S32 Appendix. Comparison-adjusted funnel plots.**
(PDF)

## Author Contributions

**Conceptualization:** Fei Wen, Liangliang Ding.

**Data curation:** Zifeng Huang, Hua Huang.

**Formal analysis:** Liangliang Ding, Yuxia Mo.

**Methodology:** Zifeng Huang, Hua Huang.

**Resources:** Fei Wen, Hua Huang.

**Software:** Liangliang Ding, Kaikai Li.

**Supervision:** Hongheng Du, Jinxi Hu.

**Validation:** Hongheng Du, Anyin Kuang.

**Visualization:** Liangliang Ding, Kaikai Li.

**Writing – original draft:** Liangliang Ding, Jinxi Hu.

**Writing – review & editing:** Hongheng Du, Anyin Kuang.

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
