## [Decision Letter · Decision Letter 0]

27 Jan 2020

PONE-D-19-32748

Clinical efficacy and safety of drug interventions for primary and secondary prevention of osteoporotic fractures in postmenopausal women: network meta-analysis followed by factor and cluster analysis

PLOS ONE

Dear Dr. Ding,

Thank you for submitting your manuscript to PLOS ONE. After careful consideration, we feel that it has merit but does not fully meet PLOS ONE’s publication criteria as it currently stands. Therefore, we invite you to submit a revised version of the manuscript that addresses the points raised during the review process.

We would appreciate receiving your revised manuscript by Mar 12 2020 11:59PM. To enhance the reproducibility of your results, we recommend that if applicable you deposit your laboratory protocols in protocols.io, where a protocol can be assigned its own identifier (DOI) such that it can be cited independently in the future. For instructions see: http://journals.plos.org/plosone/s/submission-guidelines#loc-laboratory-protocols

We look forward to receiving your revised manuscript.

Kind regards,

David Fyhrie

Academic Editor

PLOS ONE

Journal Requirements:

2. In the Methods, please specify any assessment of risk of bias that may affect the cumulative evidence (e.g., publication bias, selective reporting within studies). Please ensure that the specific method of assessment (funnel plot, Egger's test, Begg's test, etc) is mentioned.

3. Thank you for stating the following on the title page of your manuscript:

"Funding: No specific funding was received for this study."

Please remove any funding-related text from the manuscript and let us know how you would like to update your Funding Statement. Currently, your Funding Statement reads as follows: "NO"

Reviewers' comments:

Reviewer's Responses to Questions

**Comments to the Author**

1. Is the manuscript technically sound, and do the data support the conclusions?

Reviewer #1: Partly

2. Has the statistical analysis been performed appropriately and rigorously? 

Reviewer #1: I Don't Know

3. Have the authors made all data underlying the findings in their manuscript fully available?

Reviewer #1: Yes

4. Is the manuscript presented in an intelligible fashion and written in standard English?

Reviewer #1: No

5. Review Comments to the Author

Reviewer #1: The authors have made a substantial job in summarizing evidence from RCTs on drug intervention for the prevention of osteoporotic fractures.

The manuscript is rather technical and some aspects of the analysis of data can be difficult to follow. The English of this article is rather good but could be further improved by English editing.

I have some comments:

When judging the evidence (e.g. figure 5), is uncertainty taken properly into account? At the bottom of page 8, Cluster 1 and 2 are described, but these seem determined based on the estimates presented in the figure. Are the differences statistically significant?

Tolerability is defined as withdrawals due to adverse events. However, it does not take into account the severity of the adverse effect. This is acknowledged by the authors in the discussion, but should it also be taken into account when summarizing the main findings concerning safety?

Figs 2A-2H are very nice, but it is unclear how they should be judged. ‘evidence loop in the network plot’ could be defined more clearly. Is it purely the visual inspection saying that there was an evidence loop or no.

Did you consider including hip fracture as an additional endpoint. If not, this should be noted as a limitation of the study.

6. PLOS authors have the option to publish the peer review history of their article (what does this mean?). If published, this will include your full peer review and any attached files.

Reviewer #1: No

---

## [Author Response · Author response to Decision Letter 0]

1 Feb 2020

Dear Editor, 

Thank you very much for sending us the valuable comments of the reviewers on our manuscript (No.: PONE-D-19-32748; Title: Clinical efficacy and safety of drug interventions for primary and secondary prevention of osteoporotic fractures in postmenopausal women: network meta-analysis followed by factor and cluster analysis), which have helped us to improve the quality of our paper greatly. We have thoroughly revised our manuscript according to those comments. All comments have been addressed and itemized as follows.

Responses to Journal Requirements

Question 1:

Please ensure that your manuscript meets PLOS ONE's style requirements, including those for file naming. The PLOS ONE style templates can be found at http://www.plosone.org/attachments/PLOSOne_formatting_sample_main_body.pdf and http://www.plosone.org/attachments/PLOSOne_formatting_sample_title_authors_affiliations.pdf. 

Answer 

We have revised title page, main body and references of our manuscript as well as file naming of supplementary materials according to PLOS ONE's style requirements. 

Question 2:

In the Methods, please specify any assessment of risk of bias that may affect the cumulative evidence (e.g., publication bias, selective reporting within studies). Please ensure that the specific method of assessment (funnel plot, Egger's test, Begg's test, etc) is mentioned. 

Answer 

 We have performed the detection of publication bias, and have added corresponding sentences into the methods, results, and discussion section in the revised manuscript.

Question 3:

Thank you for stating the following on the title page of your manuscript:

"Funding: No specific funding was received for this study."

Please remove any funding-related text from the manuscript and let us know how you would like to update your Funding Statement. Currently, your Funding Statement reads as follows: "NO". 

Answer 

No specific funding was received for this study, and accordingly, we have not provided funding information in the Funding Statement or in any section of the manuscript.

Responses to Comments to the Author

Question 1:

Is the manuscript technically sound, and do the data support the conclusions?

Reviewer #1: Partly

Answer 

Although the process of data analyses in this paper was complex, we believe that every analysis step was logical and rigorous, which ensured the conclusions supported by the data.

Question 2:

Has the statistical analysis been performed appropriately and rigorously?

Reviewer #1: I Don't Know 

Answer 

The statistical analyses in this paper are complicated indeed. We performed network meta-analysis and calculated the SUCRA value for each treatment respectively in the primary prevention group and in the secondary prevention group at first. Then we did factor analysis based on the SUCRA values to obtain factor scores. At last, we conducted cluster analysis based on the factor scores to rank various interventions with both efficacy and safety taken into account. Although the statistical analyses are complex, we believe that they are logical and rigorous.

Question 3:

Have the authors made all data underlying the findings in their manuscript fully available?

Reviewer #1: Yes. 

Answer 

 None.

Question 4:

Is the manuscript presented in an intelligible fashion and written in standard English?

Reviewer #1: No. 

Answer 

We have made every effort to revise the manuscript in terms of English grammar, sentence structure, and spelling. If something wrong still exists in these aspects, please point out them and we will try our best to revise them again.

Question 5:

Review Comments to the Author

Reviewer #1: The authors have made a substantial job in summarizing evidence from RCTs on drug intervention for the prevention of osteoporotic fractures.

The manuscript is rather technical and some aspects of the analysis of data can be difficult to follow. The English of this article is rather good but could be further improved by English editing.

Answer: 

Thank you very much for the positive comments. 

The statistical analyses in this paper are complicated indeed. We performed network meta-analysis and calculated the SUCRA value for each treatment respectively in the primary prevention group and in the secondary prevention group at first. Then we did factor analysis based on the SUCRA values to obtain factor scores. At last, we conducted cluster analysis based on the factor scores to rank various interventions with both efficacy and safety taken into account. Although the statistical analyses are complex, we believe that they are logical and rigorous. Meanwhile, we have provided detailed methods and results in the manuscript for all data analyses to make them clear to readers. On the other hand, we have made every effort to revise the manuscript in terms of English grammar, sentence structure, and spelling. If something wrong still exists in these aspects, please point out them and we will try our best to revise them again.

Reviewer #1: I have some comments:

When judging the evidence (e.g. figure 5), is uncertainty taken properly into account? At the bottom of page 8, Cluster 1 and 2 are described, but these seem determined based on the estimates presented in the figure. Are the differences statistically significant?

Answer: 

This is a good point.

Although the cluster analysis in the paper ensured maximum intracluster similarity and minimum intercluster similarity according to the maximum of clustering gain [1], it failed to tell us whether the difference among various clusters was statistically significant or not. This point has been added into the fourth point in the limitations section of the revised manuscript. 

References

1. Jung Y, Park H, Du D, Drake B. A Decision Criterion for the Optimal Number of Clusters in Hierarchical Clustering. J GLOBAL OPTIM. 2003; 25(1):91-111. 

Reviewer #1: Tolerability is defined as withdrawals due to adverse events. However, it does not take into account the severity of the adverse effect. This is acknowledged by the authors in the discussion, but should it also be taken into account when summarizing the main findings concerning safety?

Answer: 

This is a good point.

Given that the evidence summary section in the paper mainly stated key findings and comparisons with previous studies, we have added this point into the fifth point in the limitations section.

Reviewer #1: Figs 2A-2H are very nice, but it is unclear how they should be judged. ‘evidence loop in the network plot’ could be defined more clearly. Is it purely the visual inspection saying that there was an evidence loop or no.

Answer: 

“Evidence loop in the network plot” means “closed loop in the network plot”, and that it exists in the network plot means the necessity of test for inconsistency. There is no closed loop in Fig 2D, which meant that it was not necessary to conduct test of inconsistency for the outcome of acceptability in the primary prevention group. In network plots, the size of every circle is proportional to the number of randomly assigned participants and indicates the sample size; and the width of the lines corresponds to the number of trials. Thus, those network plots also tell us: placebo was most commonly compared to other interventions in each network plot; on the contrary, the lines among active drugs were thin, and the number of those lines was limited, which suggested the lack of head-to-head trials comparing active drugs. Accordingly, we have revised the third paragraph in the results section and the legend of Fig 2. Meanwhile, the fact of the lack of head-to-head trials was clarified in the fourth point in the limitations section.

Reviewer #1: Did you consider including hip fracture as an additional endpoint. If not, this should be noted as a limitation of the study.

Answer: 

 This limitation of the paper has been added into the last point in the limitations section.

Question 6:

PLOS authors have the option to publish the peer review history of their article (what does this mean?). If published, this will include your full peer review and any attached files.

Do you want your identity to be public for this peer review? For information about this choice, including consent withdrawal, please see our Privacy Policy.

Reviewer #1: No

Answer 

None.

Hopefully, we have made an appropriate revision based on these valuable comments and suggestions. Please let me know if there is anything else we need to do with the revision, and we’ll complete it as soon as possible.

We thank you wholeheartedly for your excellent work. Your kind assistance is greatly appreciated. We look forward to any future correspondence.

Yours sincerely,

Liangliang Ding

E-mail: 15887847798m0@sina.cn

---

## [Decision Letter · Decision Letter 1]

11 May 2020

PONE-D-19-32748R1

Clinical efficacy and safety of drug interventions for primary and secondary prevention of osteoporotic fractures in postmenopausal women: network meta-analysis followed by factor and cluster analysis

PLOS ONE

Dear Dr. Ding,

Thank you for submitting your manuscript to PLOS ONE. After careful consideration, we feel that it has merit but does not fully meet PLOS ONE’s publication criteria as it currently stands. Therefore, we invite you to submit a revised version of the manuscript that addresses the points raised during the review process.

We would appreciate receiving your revised manuscript by Jun 25 2020 11:59PM. To enhance the reproducibility of your results, we recommend that if applicable you deposit your laboratory protocols in protocols.io, where a protocol can be assigned its own identifier (DOI) such that it can be cited independently in the future. For instructions see: http://journals.plos.org/plosone/s/submission-guidelines#loc-laboratory-protocols

We look forward to receiving your revised manuscript.

Kind regards,

Carlos M. Isales, M.D.

Academic Editor

PLOS ONE

Reviewers' comments:

Reviewer's Responses to Questions

**Comments to the Author**

1. If the authors have adequately addressed your comments raised in a previous round of review and you feel that this manuscript is now acceptable for publication, you may indicate that here to bypass the “Comments to the Author” section, enter your conflict of interest statement in the “Confidential to Editor” section, and submit your "Accept" recommendation.

Reviewer #1: (No Response)

Reviewer #2: (No Response)

2. Is the manuscript technically sound, and do the data support the conclusions?

Reviewer #1: Yes

Reviewer #2: Yes

3. Has the statistical analysis been performed appropriately and rigorously? 

Reviewer #1: I Don't Know

Reviewer #2: Yes

4. Have the authors made all data underlying the findings in their manuscript fully available?

Reviewer #1: Yes

Reviewer #2: Yes

5. Is the manuscript presented in an intelligible fashion and written in standard English?

Reviewer #1: Yes

Reviewer #2: Yes

6. Review Comments to the Author

Reviewer #1: Thank you for the revised version of the manuscript.

I appreciate that you have included in the limitation section that:

'Meanwhile, the two safety outcomes failed to consider the severity of adverse effects.'

(I suggest omitting 'Meanwhile')

This limitation should also be indicated in the abstract and other places where the main findings are summarized. For example, when it is stated (abstract) that: 'Romosozumab, teriparatide, denosumab and risedronate, with the greatest composite scores, constituted the optimal cluster having both superior efficacy and superior safety', the words 'superior safety is a very strong statement. Many reads do not read the limitation section.

You have also mention some other very relevant limitation. They should also influence on the rest of the discussion.

It is stated on page 11: 'Fortunately, we have identified the newest evidence on zoledronate intravenously used at a dose of 5 mg per 18 months, which nowadays is the only one drug intervention that has the capability to significantly reduce both vertebral (RR 0.46) and nonvertebral fracture risk (HR 0.66) for primary prevention of PMO'

COMMENT: As you indicate in the limitation section, this could be due to lack of studies/statistical power. I therefor suggest that you reformulate to 'which is the only drug which has been shown to significantly reduce....'

In the abstract, SUCRA should be defined

Reviewer #2: This study provided a network meta-analysis (NMA) that enabled the authors to pool data from 57 individual studies that included evidence from both direct and indirect comparisons of the 15 interventions for two efficacy outcomes and two safety outcomes.

The surface under the cumulative ranking curve (SUCRA) is a numeric presentation of the overall ranking of an intervention and presents a single number associated with each treatment. SUCRA values range from 0 to 1. Higher SUCRA values indicate that the intervention was more likely to be in the top rank while lower SUCRA values indicate that the intervention likely to be in the lower rank.

Since the SUCRA values were used to rank the interventions as well as used in the subsequent factor and cluster analyses it is important to consider their quality. There are a few problem associated with SUCRA values (1) some of which the authors have addressed.

First, the evidence on which the SUCRA rankings are based may be of very low quality and therefore untrustworthy. The authors have provided the Jadad scores along with the data corresponding to all of the selected studies. They have also provided visual representation of NMA point estimates and certainty intervals comparing estimates of each treatment against placebo in the main text as well as the comparisons among the treatments in the Supplemental files.

Second, there are typically several relevant outcomes. A treatment that is best in one outcome (say, an efficacy outcome) may be the worst in another outcome (for example, a safety outcome). The authors were able to combine efficacy and safety outcomes for the secondary prevention group via factor/cluster analyses to attempt to address this and thereby provide rankings for the interventions based on all outcomes.

There are a few problems that were not addressed and might be good to include in the Discussion or the Limitations section. Issues such as cost and a clinician’s familiarity with use of a particular treatment may also bear consideration when ranking interventions. Also, in the process of calculation, SUCRA does not consider the magnitude of differences in effects between treatments (e.g. the first ranked treatment may be only slightly, or a great deal better than the second ranked treatment). And finally, chance may explain any apparent difference between treatments, and SUCRA does not capture that possibility.

All in all it appears that all of the methods associated with this NMA are appropriate well described. The presentations of the assessments of heterogeneity and publication bias are important aids for the interpretation of the results.

(1) Mbuagbaw, L., Rochwerg, B., Jaeschke, R. et al. Approaches to interpreting and choosing the best treatments in network meta-analyses. Syst Rev 6, 79 (2017). https://doi.org/10.1186/s13643-017-0473-z

7. PLOS authors have the option to publish the peer review history of their article (what does this mean?). If published, this will include your full peer review and any attached files.

Reviewer #1: No

Reviewer #2: No

---

## [Author Response · Author response to Decision Letter 1]

13 May 2020

Dear Editor, 

Thank you very much for sending us the valuable comments of the reviewers on our manuscript (No.: PONE-D-19-32748R1; Title: Clinical efficacy and safety of drug interventions for primary and secondary prevention of osteoporotic fractures in postmenopausal women: network meta-analysis followed by factor and cluster analysis), which have helped us to improve the quality of our paper greatly. We have thoroughly revised our manuscript according to those comments. All comments have been addressed and itemized as follows.

Responses to Journal Requirements

To enhance the reproducibility of your results, we recommend that if applicable you deposit your laboratory protocols in protocols.io, where a protocol can be assigned its own identifier (DOI) such that it can be cited independently in the future. For instructions see: http://journals.plos.org/plosone/s/submission-guidelines#loc-laboratory-protocols.

Answer 

 We have deposited the study protocol for this network meta-analysis in protocols.io, and the DOI of the study protocol has been presented in the first paragraph of the METHODS section. 

Responses to Comments to the Author

Question 1:

If the authors have adequately addressed your comments raised in a previous round of review and you feel that this manuscript is now acceptable for publication, you may indicate that here to bypass the “Comments to the Author” section, enter your conflict of interest statement in the “Confidential to Editor” section, and submit your "Accept" recommendation.

Reviewer #1: (No Response)

Reviewer #2: (No Response)

Answer 

None.

Question 2:

Is the manuscript technically sound, and do the data support the conclusions?

Reviewer #1: Yes

Reviewer #2: Yes 

Answer 

Thanks very much for these positive responses.

Question 3:

Has the statistical analysis been performed appropriately and rigorously?

Reviewer #1: I Don't Know

Reviewer #2: Yes. 

Answer 

 This point has been detailed in the first response letter, and we believe the statistical analyses in this paper were logical and rigorous although they were complicated. Thanks very much for the positive response of Reviewer #2.

Question 4:

Have the authors made all data underlying the findings in their manuscript fully available?

Reviewer #1: Yes

Reviewer #2: Yes. 

Answer 

Thanks very much for these positive responses.

Question 5:

Is the manuscript presented in an intelligible fashion and written in standard English?

Reviewer #1: Yes

Reviewer #2: Yes. 

Answer 

Thanks very much for these positive responses.

Question 6:

Review Comments to the Author

Please use the space provided to explain your answers to the questions above. You may also include additional comments for the author, including concerns about dual publication, research ethics, or publication ethics. (Please upload your review as an attachment if it exceeds 20,000 characters).

Reviewer #1: Thank you for the revised version of the manuscript.

I appreciate that you have included in the limitation section that:

'Meanwhile, the two safety outcomes failed to consider the severity of adverse effects.'

(I suggest omitting 'Meanwhile')

This limitation should also be indicated in the abstract and other places where the main findings are summarized. For example, when it is stated (abstract) that: 'Romosozumab, teriparatide, denosumab and risedronate, with the greatest composite scores, constituted the optimal cluster having both superior efficacy and superior safety', the words 'superior safety is a very strong statement. Many reads do not read the limitation section.

Answer: 

We have omitted 'Meanwhile' in the sentence of 'Meanwhile, the two safety outcomes failed to consider the severity of adverse effects'.

We have added this limitation into the ABSTRACT section and into the second paragraph of the DISCUSSION section.

Reviewer #1: You have also mention some other very relevant limitation. They should also influence on the rest of the discussion.

It is stated on page 11: 'Fortunately, we have identified the newest evidence on zoledronate intravenously used at a dose of 5 mg per 18 months, which nowadays is the only one drug intervention that has the capability to significantly reduce both vertebral (RR 0.46) and nonvertebral fracture risk (HR 0.66) for primary prevention of PMO'

COMMENT: As you indicate in the limitation section, this could be due to lack of studies/statistical power. I therefor suggest that you reformulate to 'which is the only drug which has been shown to significantly reduce....'.

Answer: 

 We have revised this sentence. Meanwhile, we have revised the sentences relevant with this point in the ABSTRACT and CONCLUSIONS sections.

Reviewer #1: In the abstract, SUCRA should be defined.

Answer: 

SUCRA has been defined in the ABSTRACT section.

Reviewer #2: This study provided a network meta-analysis (NMA) that enabled the authors to pool data from 57 individual studies that included evidence from both direct and indirect comparisons of the 15 interventions for two efficacy outcomes and two safety outcomes.

The surface under the cumulative ranking curve (SUCRA) is a numeric presentation of the overall ranking of an intervention and presents a single number associated with each treatment. SUCRA values range from 0 to 1. Higher SUCRA values indicate that the intervention was more likely to be in the top rank while lower SUCRA values indicate that the intervention likely to be in the lower rank.

Since the SUCRA values were used to rank the interventions as well as used in the subsequent factor and cluster analyses it is important to consider their quality. There are a few problem associated with SUCRA values (1) some of which the authors have addressed.

(1) Mbuagbaw, L., Rochwerg, B., Jaeschke, R. et al. Approaches to interpreting and choosing the best treatments in network meta-analyses. Syst Rev 6, 79 (2017). https://doi.org/10.1186/s13643-017-0473-z

First, the evidence on which the SUCRA rankings are based may be of very low quality and therefore untrustworthy. The authors have provided the Jadad scores along with the data corresponding to all of the selected studies. They have also provided visual representation of NMA point estimates and certainty intervals comparing estimates of each treatment against placebo in the main text as well as the comparisons among the treatments in the Supplemental files.

Second, there are typically several relevant outcomes. A treatment that is best in one outcome (say, an efficacy outcome) may be the worst in another outcome (for example, a safety outcome). The authors were able to combine efficacy and safety outcomes for the secondary prevention group via factor/cluster analyses to attempt to address this and thereby provide rankings for the interventions based on all outcomes.

All in all it appears that all of the methods associated with this NMA are appropriate well described. The presentations of the assessments of heterogeneity and publication bias are important aids for the interpretation of the results.

Answer: 

Thanks very much for these positive comments.

Reviewer #2: There are a few problems that were not addressed and might be good to include in the Discussion or the Limitations section. Issues such as cost and a clinician’s familiarity with use of a particular treatment may also bear consideration when ranking interventions. Also, in the process of calculation, SUCRA does not consider the magnitude of differences in effects between treatments (e.g. the first ranked treatment may be only slightly, or a great deal better than the second ranked treatment). And finally, chance may explain any apparent difference between treatments, and SUCRA does not capture that possibility.

Answer: 

 The first point has been added into the last paragraph of the Clinical Implications section in the DISCUSSION section. The second and third points have been added into the last point of the Limitations section.

Question 7:

PLOS authors have the option to publish the peer review history of their article (what does this mean?). If published, this will include your full peer review and any attached files.

Do you want your identity to be public for this peer review? For information about this choice, including consent withdrawal, please see our Privacy Policy.

Reviewer #1: No

Reviewer #2: No

Answer 

None.

Hopefully, we have made an appropriate revision based on these valuable comments and suggestions. Please let me know if there is anything else we need to do with the revision, and we’ll complete it as soon as possible.

We thank you wholeheartedly for your excellent work. Your kind assistance is greatly appreciated. We look forward to any future correspondence.

Yours sincerely,

Liangliang Ding

E-mail: 15887847798m0@sina.cn

---

## [Editor Report · Decision Letter 2]

20 May 2020

Clinical efficacy and safety of drug interventions for primary and secondary prevention of osteoporotic fractures in postmenopausal women: network meta-analysis followed by factor and cluster analysis

PONE-D-19-32748R2

Dear Dr. Ding,

We are pleased to inform you that your manuscript has been judged scientifically suitable for publication and will be formally accepted for publication once it complies with all outstanding technical requirements.

With kind regards,

Carlos M. Isales, M.D.

Academic Editor

PLOS ONE
---

## [Editor Report · Acceptance letter]

22 May 2020

PONE-D-19-32748R2 

Clinical efficacy and safety of drug interventions for primary and secondary prevention of osteoporotic fractures in postmenopausal women: network meta-analysis followed by factor and cluster analysis 

Dear Dr. Ding:

I am pleased to inform you that your manuscript has been deemed suitable for publication in PLOS ONE. Congratulations! Your manuscript is now with our production department. 

With kind regards,

on behalf of

Professor Carlos M. Isales 

Academic Editor

PLOS ONE